# Revision Surgery for Shoulder Infection after Arthroscopic Rotator Cuff Repair: Functional Outcomes and Eradication Rate—A Systematic Review

**DOI:** 10.3390/healthcare12131291

**Published:** 2024-06-28

**Authors:** Michele Mercurio, Giorgio Gasparini, Erminia Cofano, Stefano Colace, Olimpio Galasso

**Affiliations:** 1Department of Orthopaedic and Trauma Surgery, Magna Graecia University, 88100 Catanzaro, Italy; michele.mercurio@unicz.it (M.M.); gasparini@unicz.it (G.G.); stefanoleonardo.colace@studenti.unicz.it (S.C.); galasso@unicz.it (O.G.); 2Research Center on Musculoskeletal Health, MusculoSkeletal Health@UMG, Magna Graecia University, 88100 Catanzaro, Italy; 3Department of Medicine, Surgery and Dentistry, University of Salerno, 84081 Baronissi, Italy

**Keywords:** rotator cuff repair, shoulder infection, revision surgery, shoulder arthroscopy, cutibacterium acnes

## Abstract

The outcomes after arthroscopic rotator cuff repair (RCR) have been reported to be successful. The incidence of deep infections (defined as an infection involving any part of the anatomy other than the skin and subcutaneous tissue) after surgery ranges between 0.03% and 3.4%. This systematic review aims to investigate the outcomes of revision surgery for infection following arthroscopic RCR. Clinical outcomes and eradication rates among patients treated with different surgical and antibiotic therapies are analyzed. A total of five studies were eligible for systematic review. A total of 146 patients were treated and evaluated, of whom 71 (48%) and 75 (52%) underwent arthroscopic and open surgery to manage the infection, respectively. The most common causative bacterium was *Cutibacterium acnes* (50.4%). Two studies reported the pre-and postoperative ASES score and Constant–Murley score (CMS), and a statistically significant improvement was found after surgery (*p* < 0.001 for both). Eradication was observed in a total of 138 patients (94.5%); no difference was found between arthroscopic and open revision surgery (92.8% and 96%, respectively, *p* = 0.90). The frequency-weighted mean duration of the intravenous antibiotic therapy was 6.6 ± 5.4 days, while the overall mean duration of antibiotic therapy, considering intravenous and oral administration, was 43.5 ± 40 days. Patients with infection following arthroscopic RCR undergoing revision surgery experienced a high rate of eradication. A significant improvement in shoulder functionality and less residual pain can be expected.

## 1. Introduction

A common cause of shoulder pain and disability is rotator cuff injuries. Non-operative treatment and surgical repair are the management options for rotator cuff injuries. A meta-analysis found no significant differences in clinical outcomes, pain, or operative time between arthroscopic and mini-open rotator cuff repair (RCR) despite the variety of surgical options [1]. Shoulder arthroscopy has become a widespread surgical technique for treating various pathologies of the shoulder. Good-to-excellent results have been reported after arthroscopic rotator cuff repair, even in elderly populations [2]. The advantages of arthroscopic repair are reduced deltoid trauma, reduced risk of axillary nerve injury, and improved intraoperative visualization, tendon mobilization, and cosmesis [3]. Complications of RCR include postoperative shoulder stiffness, anchor pull-out, RCR failure, deep vein thrombosis, and postoperative infection. The overall complication rate after the index procedure ranges from 4.8% to 10.6%, and several risk factors have been described in the literature, such as male sex, duration of surgery greater than 90 min, rheumatoid arthritis, and intravenous drug use [4]. The incidence of deep infections has been previously reported to be between 0.03% and 3.4%, representing a potentially devastating postoperative complication [5]. A Delphi consensus process on complications associated with arthroscopic RCR defined the deep infection as an infection involving any part of the anatomy (e.g., fascia, muscle, organs, and spaces) other than the skin and subcutaneous tissue of the incision [6]. The diagnosis of postoperative deep infections usually includes the following criteria: drainage persisting longer than five days since index operation, the development of a sinus tract to the joint, two or more positive cultures at the revision surgery with the same organisms, or the presence of purulence [7]. The occurrence of infections after shoulder arthroscopy can lead to prolonged hospital stays, increased morbidity, and compromised surgical outcomes [8].

The current literature on infections after arthroscopic RCR is limited by small sample sizes and variable [9] treatments, including intravenous or oral antibiotics, single or multiple debridement with open or arthroscopic surgery, and a combination of the above [7]. The removal of anchors and sutures is also disputed. Several factors, including the patient’s age, postsurgical expectations [10], and virulence of the infectious agent influence the choice of treatment, and the gold standard to manage infection after RCR is still under debate.

The goal of this systematic review is to analyze the clinical outcomes and eradication rates among patients treated with different revision surgery and antibiotic therapies for infection following arthroscopic RCR.

## 2. Materials and Methods

### 2.1. Search Strategy

According to the Preferred Reporting Items for Systematic Reviews and Meta-Analyses (PRISMA) statement [11], a systematic review of the published literature was conducted and reported. The study protocol was registered prospectively in PROSPERO (CRD525387). The PubMed, MEDLINE, Scopus, and Cochrane Central databases were searched for articles published until January 2024. The search terms used to retrieve relevant articles were “arthroscopy”, “infection”, “rotator cuff repair”, “results”, “outcome”, “eradication”, and “complications” [Appendix A]. Two authors (MM and SC) independently reviewed the titles and abstracts and contacted a third senior author (OG) if there were significant discrepancies. The reference list for each included article, as well as the available grey literature at our institution, was screened for potential additional articles. We did identify other articles with this additional research.

### 2.2. Inclusion Criteria and Study Selection

The following inclusion criteria were applied during title, abstract, and full-text screening according to the PICO format [12]: (1) Population: patients with early and late infection after shoulder arthroscopy for rotator cuff repair; (2) Intervention: studies reporting > 5 surgical cases, treated with open or arthroscopic surgery to manage infection; (3) Comparator: regardless of the presence or absence of comparator or control groups, all studies were eligible (4) Outcome: articles written in English, reporting outcomes and/or eradication rates with a minimum mean follow-up of 12 months. Other reviews, case reports, articles without outcomes or results, cadaveric or biomechanical studies, technical notes, editorials, letters to the editor, and expert opinions were excluded from the analysis but considered for the discussion section.

### 2.3. Data Extraction and Quality Assessment

The revision of the included studies and the extraction of the data were performed by two surgeons (MM and SC). For each article, the first author, journal name, year of publication, level of evidence, and patient demographics were recorded. The data extracted for quantitative analysis included the timing of the shoulder infection and the visual analogue scale (VAS) score for pain, ranging from 0 (no pain) to 10 (extreme pain) [10]. The American Shoulder and Elbow Surgeons (ASES) score contains a physician-rated and patient-rated score, evaluation of pain, and 10 functional questions [2]. The Constant–Murley score (CMS) comprises measures of capacity (ROM and strength) and subjective parameters such as workload, pain, and leisure time activities with a score ranging from 0 (worst) to 100 (best) [2]. Microbiology, the duration of the antibiotic therapy, and the eradication rate were also analyzed.

A methodological quality assessment using the Modified Newcastle–Ottawa Quality Assessment Scale [13] was independently performed by 3 authors (MM, EC, and SC). Disagreements were resolved by consultation with a senior reviewer (OG) with over 20 years of experience in the field of shoulder surgery. The details of the quality assessment are shown in Table 1.

### 2.4. Statistical Analysis

The quantitative data were organized for statistical analysis; all the data were collected, measured, and reported with 1-decimal accuracy. Weighted means and standard deviations were calculated for data concerning demographic characteristics and outcomes. When standard deviations were not directly provided, they were calculated with the equation [max range − min range/4] to allow for statistical aggregation [15]. The weighted mean and standard deviation comparisons were performed using unpaired *t*-tests, and 2 × 2 contingency tables were used to compare proportions. All the tests were performed with SPSS Statistics software (version 25.0; IBM Corp., Armonk, NY, USA) and GraphPad Prism (version 7.0; GraphPad Software Inc., San Diego, CA, USA). The 95% confidence intervals were calculated, and a *p*-value less than 0.05 was considered to indicate statistical significance.

## 3. Results

In total, the initial search identified 625 relevant articles; 390 abstracts were screened, and 40 full-text articles were assessed for eligibility based on our inclusion criteria. This resulted in five studies that were eligible for systematic review (Figure 1).

The baseline characteristics of these studies are summarized in Table 2.

A total of 146 patients were treated and evaluated, of whom 71 (48%) and 75 (52%) underwent arthroscopic and open surgery to manage the infection, respectively. Overall, 89.7% of the patients were male. The frequency-weighted mean age at the time of the operation was 60.4 ± 9.5 years, and the frequency-weighted mean at follow-up was 77 ± 42.6 months. The prevalence of diabetes and smoking habits were 21% and 17.1%, respectively.

### 3.1. Microbiology

The causative pathogens are reported in Table 3.

Data for perioperative cultured pathogens were reported for 143 cultures, and the most common causative bacteria were *Cutibacterium acnes* (46.2%), *Staphylococcus epidermidis* (17.5%), and *coagulase-negative Staphylococcus (CNS)* (13.3%). Cultures with no pathogens isolated amounted to 9.2% of the total. A total of 10 patients had multiple causative pathogens.

### 3.2. Clinical and Functional Outcomes

The pre-and postoperative clinical outcome data are shown in Table 4.

One study [4] reported the pre- and postoperative VAS scores in 30 patients, with mean values of 7.1 ± 1.3 and 1.7 ± 2.5, respectively (*p* < 0.001). Two studies [4,7] with 53 patients reported the pre- and postoperative ASES scores, and a statistically significant improvement was found after surgery (35 ± 11.1 vs. 74.3 ± 19.7, respectively, *p* < 0.001). Two studies [4,7] with a total of 41 patients reported the pre- and postoperative CMSs and a statistically significant improvement was found at follow-up (27.3 ± 12.3 vs. 69.7 ± 25.6, respectively, *p* < 0.001). Before surgery, the mean CMS was 30% of that of sex- and age-matched healthy individuals [16]. The mean postoperative CMS was 76% of that of sex- and age-matched healthy individuals. The mean increase in the CMS was 42 points.

### 3.3. Treatment Options and Eradication Rate

A total of 138 patients (94.5%) experienced eradication (Table 5).

A total of 65 out of 70 (92.8%) and 72 out of 75 (96%) patients who underwent arthroscopic or open revision surgery experienced eradication, respectively (*p* = 0.90). A total of 110 sutures and anchors in 123 patients were removed.

Data on the duration of antibiotic treatment and the type of surgery performed to manage the infection are reported in Table 6.

All 146 patients received postoperative antibiotic therapy. The frequency-weighted mean duration of the intravenous antibiotic therapy was 6.6 ± 5.4 days, while the overall frequency-weighted mean duration of antibiotic therapy, considering intravenous and oral administration, was 43.5 ± 40 days.

## 4. Discussion

In the present study, the most common causative pathogen causing shoulder infection following arthroscopic RCR was *C. acnes*. The eradication was reported in 94.5% of patients with no difference between arthroscopic and open revision surgery adopted to manage the infection. Patients undergoing revision surgery showed significant functional improvement and less residual pain as measured by the ASES score, CMS, and VAS score. The patients received intravenous and oral antibiotic therapy for a mean duration of 6 weeks.

This study updates the current evidence from previous systematic reviews on this topic. Indeed, these are mostly descriptive in nature and aggregate data from heterogeneous studies, focusing on the overall complication rates without reporting specific data on the management and outcomes of revision surgery after shoulder infection following arthroscopic RCR.

RCR by arthroscopy is currently the “gold standard” treatment to minimize peri-operative morbidity and accelerate functional recovery, and complication rate analysis confirmed that arthroscopy is a low-risk surgical procedure [17] compared to open surgery, which is associated with a higher risk of shoulder infection [18]. The portals in the arthroscopic RCR are smaller and leave the joint less exposed, reducing the risk of infection [4]. A study by Huges et al. [19] investigated the infection rates between arthroscopic and open RCR and showed that open surgery was 5.6 times more likely to develop a postoperative infection. Several risk factors for postoperative infection after arthroscopic RCR were assessed in the literature. Agarwalla et al. [20] investigated the operation time and an increase of fifteen minutes was found to be correlated with a higher surgical site infection rate. The role of preoperative antibiotic prophylaxis was also debated. Pauzenberger et al. [5] showed a reduction in the incidence of postoperative infections from 1.6% to 0.3% after introducing prophylactic antibiotics, while Baraza et al. [21] found that the absence of prophylactic antibiotics in low-risk patients do not increase the infection risk. Patients with deep infections following arthroscopic RCR present with pain, stiffness, and in some cases, a loss of postoperative range of motion in the shoulder. Fibrinous exudate, pus drainage from the surgical wounds, local swelling, erythema, and warmth can also be present. Systemic signs and symptoms such as fever, chills, and generalized fatigue may occur if there is delayed presentation in immunosuppressed patients. Laboratory tests are required to verify the infection. These include erythrocyte sedimentation rate (ESR) and peripheral white blood cell (WBC) count with differential and C-reactive protein (CRP) levels. ESR and CRP levels are sensitive but not specific markers. If shoulder infection is suspected, glenohumeral joint aspiration and cell count analysis should be performed routinely [22]. Negative joint aspirate cultures usually indicate that there is no infection. Unfortunately, it is not possible to completely rule out the diagnosis, as has been suggested by Bauer et al. [23]. In this regard, Athwal et al. [24] stated that the slow growth of the organism often necessitates an extended culture of at least 7 days for isolation and recommended that cultures be monitored for at least 10 days to ensure a negative culture.

Other studies investigated epidemiological data and comorbidities and their correlation with postoperative infection [18]. In the current study, the frequency-weighted mean age at the time of the operation was 60 years. Yeranosian et al. [25] reported that the infection rate after arthroscopic shoulder surgery was higher in patients over 60 years of age, with an incidence twice as often as in patients under 40 years of age; the authors suggested that these findings may have been due to age-related comorbidities and the compromised immune status of the patients making them more susceptible to infection. Vopat et al. demonstrated that male sex is a significant risk factor for developing a postoperative infection [3], and this finding concurs with those reported in our study, where 90% of the patients were men. A higher bacterial load of *C. acnes* was suggested for the shoulder area of males than for other areas of the body, possibly due to gender [26,27]. It is interesting to note that the prevalence of *C. acnes* in the current study was 50%, making it the most common causative pathogen. *C. acnes* is recognized as a skin commensal with a low virulence. Therefore, a limited local inflammatory response can be expected with serological markers, often within normal ranges [14,27]. It is noteworthy that a high prevalence of *C. acnes* is also frequently reported in periprosthetic shoulder infections, whereas this pathogen is rarely responsible for infection in other joints [27]. We also found that cultures with no pathogens isolated amounted to 9.2% of the total. The effect of irrigation fluid as part of the arthroscopic revision surgery has been suggested to dilute bacterial counts [3].

A study conducted by the American College of Surgeons National Surgical Quality Improvement Program also showed that diabetes increases the risk of infection after non-arthroplasty shoulder surgery [28]. Travern et al. [29] reported that patients with insulin-dependent diabetes have a high risk of infection after shoulder arthroscopy. Accordingly, Cerri-Droz et al. [30] found that patients with non-insulin-dependent diabetes have a high risk for sepsis after arthroscopic RCR. In the current study, 14.4% of the patients had diabetes. A similar rate of diabetes (i.e., 18%) among patients with complications after RCR was reported by Owens et al. [31]. Interestingly, the prevalence of diabetes mellitus in the general population is 5.9% [32], and our data support its pathogenetic role.

We also found a significant improvement after revision surgery in the ASES score and CMS at the follow-up. Our results concur with those reported by Athwal et al. [24], which showed a significant improvement in the ASES score after revision surgery. Stone et al. [7] reported a significant increase in the ASES score, with no differences in functional outcomes between patients who underwent arthroscopic or open revision surgery; the authors also reported that postoperative functional scores were lower than those expected after primary RCR surgery. Similarly, in the current study, the mean postoperative CMS was 76% of that of sex- and age-matched healthy individuals. However, the mean increase of 42 points in the CMS was greater than the minimal clinically important difference for the CMS (10.4 points) [33].

In the current study, eradication was reported in 94.5% of patients, with no difference between arthroscopic and open revision surgery adopted to manage the infection. It should be considered that patients who undergo arthroscopic revision for infection after RCR often need repeated debridement until eradication [14]. Stone et al. [7] reported that open debridement also resulted in the need for repeated debridement to adequately treat infection after RCR; the authors suggested that the infections requiring open debridement were more severe than those requiring arthroscopy. In the case of open debridement, the risks correlated to general anesthesia and higher invasiveness should be also discussed with the patient [4]. The removal of sutures and anchors is controversial. Atwhal et al. [24] recommended keeping the implant, showing better outcomes after debridement. Similarly, Mirazayan et al. [34] stated that devices can be retained if the infection is not extensive and if there is no retear of the repaired cuff in order to avoid impacting the success of the surgery. On the contrary, Peuzenberger et al. [5] removed all the sutures and anchors from the infected shoulder. The removal of anchors and sutures did not affect shoulder function after infection, as reported by Jenssen et al. [14]. However, the insufficient remaining rotator cuff tissue does not always allow for new RCR at the time of revision surgery. Hartzler et al. [35] suggested avoiding the routine removal of the suture anchor but only if it was broken during the revision surgery because this compromised the remaining soft tissue and greater tuberosity bone stock; the authors reported that the removal of anchors may preclude the possibility of future RCR or increase the complexity of any subsequent revision procedure. Ammann et al. [8] found that the removal of anchors or sutures, repeated reoperation, or antibiotic therapy beyond 6 weeks did not enhance remission or decrease sequelae; these findings are a practical conclusion, in line with the literature, and the author’s recommendation is to avoid the simultaneous removal of the implant devices unless they are loose and easily accessible via surgery.

In this context, there is no consensus about the duration of antibiotic therapy, and an overall mean duration of 6 weeks considering intravenous and oral administration was reported in the current study. The management of infection commonly includes 4–6 weeks of intravenous antibiotics, and the OVIVA non-inferiority trial found that oral antibiotic therapy is non-inferior to intravenous therapy in the treatment of bone and joint infection [36]. Microbiological culture and susceptibility results of the causative microorganism should guide the choice of antibiotics. Based on the expected findings of *C. acnes* and Staphylococcus species according to the Infection Diseases Society of America guideline for the treatment of *C. acnes* for prosthetic joint infection, Jenssen et al. [14] suggested initial empiric antibiotic treatment with Vancomycin 1 g every 12 h and Penicillin 1.3 g every 6 h s. Atesok et al. [22] proposed that the initial intravenous antibiotic treatment, performed for 4 to 6 weeks, can be supplemented or prolonged by oral antibiotics depending on the causative pathogen and the patient’s response to ensure the eradication of the infection. The assessment of therapeutic response and follow-up is mainly based on the patient’s clinic and monitoring of infection markers. The duration of antibiotic treatment and confirmation of the eradication of infection requires shared decision-making between infectious disease specialists and orthopedic surgeons. The results of this study should be interpreted with caution because it has several limitations. First, we excluded technical notes and case reports because the inclusion of articles with a higher risk of bias could disqualify the systematic review. Indeed, it was reported that case reports that typically include up to four patients are limited by their retrospective and nonblinded design, constituting a source of bias that could affect the study outcome. Moreover, the findings provided by case reports may not be generalizable or useful for establishing a cause–effect relationship, as reported by Sampayo-Cordero et al. [37]. Second, only studies published in the English language were included, which may have contributed to publication bias; furthermore, this search involved four major literature databases; so, we cannot exclude the possibility that additional articles could have been identified using other databases. Third, heterogeneity in terms of sample size was found between the included studies. Fourth, it was not possible to compare outcomes according to the type of revision surgery used (arthroscopic or open), and there was heterogeneity in terms of the treatment of infection between the included articles that did not provide details about the surgical technique used to manage the infection. Fifth, it was not possible to compare the duration and type of the administration of antibiotic therapy. Finally, we found heterogeneity in the mean follow-up time of the included studies. It is likely that eradication rates are influenced by the length of the evaluation periods. Similarly, if a specific and longer follow-up time is used, the results could potentially differ between the procedures. Clinicians should not ignore the fact that differences in patients’ characteristics may favor one treatment option over another and that appropriate patient selection and surgical techniques are critical to maximizing outcomes when searching for the best surgical procedures to treat rotator cuff infection [38].

## 5. Conclusions

*C. acnes* is the most common causative pathogen causing shoulder infection following arthroscopic RCR. Eradication was reported in 94.5% of patients, with no difference between arthroscopic and open revision surgery adopted to manage the infection. All patients received antibiotic therapy for a mean duration of 6 weeks. Patients undergoing revision surgery showed significant improvement in shoulder functionality and less residual pain. The results of the current study may be of interest to patients, clinicians, and policymakers in the orthopedic field. They may help to guide evidence-based decision-making, ultimately improving patient outcomes and awareness of the challenging management of infection after RCR.

## Figures and Tables

**Figure 1 healthcare-12-01291-f001:**
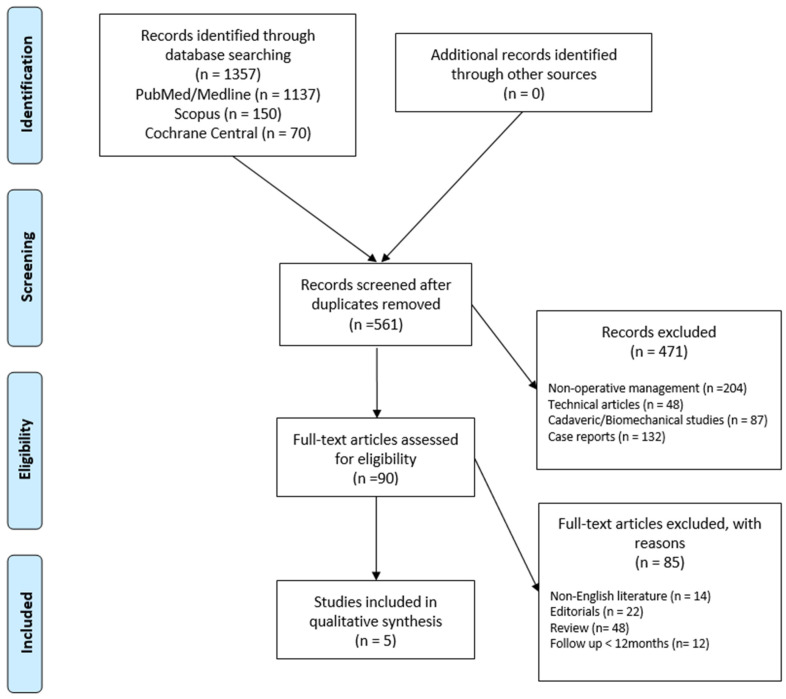
Preferred reporting items for systematic review and meta-analysis (PRISMA) flowchart for the search and identification of included studies.

**Table 1 healthcare-12-01291-t001:** Newcastle–Ottawa Scale.

Study Author (Year)	Criteria								Total	Quality
	1	2	3	4	5	6	7	8		
Ammann E et al. (2020) [8]	1	0	1	1	2	1	1	1	8	High
Frank JK et al. (2020) [4]	1	0	1	1	2	1	1	1	8	High
Jenssen KK et al. (2018) [14]	1	0	1	1	2	1	1	1	8	High
Pauzenberger L et al. (2017) [5]	1	0	1	1	2	1	1	1	9	High
Stone MA et al. (2023) [7]	1	0	1	1	2	1	1	1	8	High

Based on the total score, quality was classified as “low” (0–3), “moderate” (4–6), and “high” (7–9). Criterion number (in bold): 1, representativeness of the exposed cohort; 2, selection of the nonexposed cohort; 3, ascertainment of exposure; 4, demonstration that the outcome of interest was not present at the start of this study; 5, comparability of cohorts on the basis of the design or analysis; 6, assessment of the outcome; 7, was follow-up long enough for outcomes to occur?; 8, adequacy of the follow-up of the cohorts. Each study was awarded a maximum of one or two points for each numbered item within categories based on the Modified Newcastle–Ottawa scale rules.

**Table 2 healthcare-12-01291-t002:** Characteristics of included studies.

Author	Journal	Year of Publication	Scientific Level	Years of Study	FU (Months)			Patients’ Demographics					
								Initial Patients	Gender		Age (Years)		
					Mean	SD	Range	N	M	F	Mean	SD	Range
Jenssen KK et al. [14]	Knee Surgery, Sports, Traumatology, Arthroscopy	2018	III	2010–2014	24	4.25	11–28	11	11	0	56	82	41–68
Frank JK et al. [4]	Arthroscopy, Sports Medicine, and Rehabilitation	2020	IV	2002–2016	100	30	48–168	30	28	2	62.7	10.4	38–79
Pauzenberger L et al. [5]	Knee Surgery, Sports, Traumatology, Arthroscopy	2017	III	2004–2014	120	NA	NA	28	27	1	64.2	8.2	NA
Stone MA et al. [7]	The Archives of Bone and Joint Surgery	2023	IV	2003–2017	72.4	43.2	14.5–195.8	23	21	2	55	7	38–73
Ammann E et al. [8]	Journal of Shoulder and Elbow Surgery	2020	III	1999–2017	24	NA	NA	54	44	10	54	NA	NA
Total								146	131	15			

FU: follow-up, SD: standard deviation, N: number of patients, and NA: not available.

**Table 3 healthcare-12-01291-t003:** Causative pathogens in shoulder infections after arthroscopic rotator cuff repair.

Pathogens	No.	%
*Cutibacterium acnes*	66	46.2%
*Staphylococcus epidermidis*	25	17.5%
*coagulase-negative Staphylococcus*	19	13.3%
*Methicillin-Sensitive Staphylococcus aureus*	4	2.8%
*Gram-positive cocci (not otherwise specified)*	4	2.8%
*Staphylococcus hominis*	3	2.1%
*Methicillin-Resistant Staphylococcus epidermidis*	2	1.4%
*Staphylococcus capitis*	2	1.4%
*Actinomyces*	2	1.4%
*Staphylococcus w* *arneri*	1	0.7%
*Streptococcus agalactiae*	1	0.7%
*Pseudomonas aeruginosa*	1	0.7%
*Absidia species*	1	0.7%
*No pathogens isolated*	12	9.2%
Total	143	100.0%

**Table 4 healthcare-12-01291-t004:** The pre-and postoperative clinical outcome data.

Authors	Patients	VAS Preoperative		VAS Postoperative		ASES Preoperative		ASES Postoperative		CMS Preoperative		CMS Postoperative	
	No.	Mean	SD	Mean	SD	Mean	SD	Mean	SD	Mean	SD	Mean	SD
Jenssen KK et al. [14]	11	NA	NA	NA	NA	NA	NA	NA	NA	34	10.8	84	17.7
Frank JK et al. [4]	30	7.1	1.3	1.7	2.5	37.8	8.5	76.7	16.3	24.8	12	64.5	26.2
Pauzenberger L et al. [5]	28	NA	NA	NA	NA	NA	NA	NA	NA	NA	NA	NA	NA
Stone MA et al. [7]	23	NA	NA	NA	NA	31.3	13.1	71.1	23.4	NA	NA	NA	NA
Ammann E et al. [8]	54	NA	NA	NA	NA	NA	NA	NA	NA	NA	NA	NA	NA
Total	146												

VAS: visual analogue scale; SD: standard deviation; ASES: American Shoulder and Elbow Surgeons; CMS: Constant–Murley score; and NA: not available.

**Table 5 healthcare-12-01291-t005:** Eradication rates after revision surgery.

Authors	Arthroscopic Revision Surgery			Open Revision Surgery		
	Cases	Eradication	%	Cases	Eradication	%
	N	N		N	N	
Jenssen KK et al. [14]	11	11	100.0%	NA	NA	NA
Frank JK et al. [4]	4	4	100.0%	26	26	100.0%
Pauzenberger L et al. [5]	2	2	100.0%	26	26	100.0%
Stone MA et al. [7]	9	9	100.0%	13	13	100.0%
Ammann E et al. [8]	44	39	72.2%	10	7	70.0%
Total	70	65	92.8%	75	72	96.0%

N: Number and NA: not available.

**Table 6 healthcare-12-01291-t006:** Duration of antibiotic treatment and type of surgery performed to manage the infection.

Authors	Patients Who Underwent Antibiotic Therapy	Infection Treatment								
		Duration of Antibiotic Therapy (Days)						Surgery		N of Implants Removed
		Intravenous			Oral			Arthroscopic Revision Surgery	Open Revision Surgery	
		Mean	SD	Range	Mean	SD	Range			
Jenssen KK et al. [14]	11	15	5.3	7–28	25	7	7–32	11	0	2
Frank JK et al. [4]	30	4.5	3.8	1–16	10	21.5	4–90	4	26	4
Pauzenberger L et al. [5]	28	5.6	3.7		14.2	9.5		2	26	66
Stone MA et al. [7]	23	38 ± 10 (14–56)						10	13	NA
Ammann E et al. [8]	54	75 ± 45.5 (14–196)						44	10	38

N: Number and NA: not available.

## Data Availability

Not applicable.

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
