# Peer review of "Revision Surgery for Shoulder Infection after Arthroscopic Rotator Cuff Repair: Functional Outcomes and Eradication Rate—A Systematic Review"

_healthcare, 2024, doi:10.3390/healthcare12131291_

Round 1

Reviewer 1 Report

Comments and Suggestions for Authors

This is a very interesting field, but I felt the results part is a little confusing.

First, Table 2 is hard to read. And there are some omissions and repetitions of sentences.

Line 32-35. Rotator Cuff Repair (RCR) injuries are a common cause of shoulder pain and disability.

Rotator Cuff injuries are a common cause of shoulder pain and disability.

Line 45-48. The incidence of deep infections has been previously reported between 0.03% and 3.4% representing a potentially devastating postoperative complication while the incidence of deep infections has been 47 previously reported between 0.03% and 3.4% [5].

The incidence of deep infections has been previously reported between 0.03% and 3.4% representing a potentially devastating postoperative complication[5].

Line 153-155. “Cultures with no pathogens isolated amounted to 9.2% of the total. A total of 10 patients had multiple causative pathogens.” The numbers don't add up. How many patients who don’t reveal the cultures?

Line 289-290. “Ammann et al[8]. found that the removal of anchors or sutures, repeated revision surgery, or antibiotic therapy beyond 6 weeks failed to improve remission or to reduce sequelae.” Why? Please elaborate more detail.

Line 302-304. “Atesok et al. [21] suggested that the initial intravenous antibiotic treatment can be then supplemented or extended by oral antibiotics depending on the causative microorganism and the patient’s response.” Why? Please elaborate more detail.

Comments on the Quality of English Language

There are some omissions and repetitions of sentences. Please check more carefully.

Author Response

May 28th, 2024

Rahman Shiri                      

Editor-in-Chief

Healthcare

Manuscript ID Healthcare-2973045

Title: Revision surgery for shoulder infection after arthroscopic rotator cuff repair: functional outcomes and eradication rate – a systematic review

Dear Editor,

Enclosed please find our point-to-point response to the comments concerning our manuscript entitled “Revision surgery for shoulder infection after arthroscopic rotator cuff repair: functional outcomes and eradication rate – a systematic review” that is being considered for publication in your Journal.

Changes were made with word track changes. We hope that the revision improves our manuscript and that the work will be judged worthy of publication in your Journal.

Sincerely,

The Authors

Reviewer #1

This is a very interesting field, but I felt the results part is a little confusing.

A: Thanks for your comment. We revised the text point-to-point according to your valuable suggestions.

First, Table 2 is hard to read. And there are some omissions and repetitions of sentences.

A: Thanks for your comment. We revised the width of the columns of table 2.

Line 32-35. Rotator Cuff Repair (RCR) injuries are a common cause of shoulder pain and disability.

→Rotator Cuff injuries are a common cause of shoulder pain and disability.

A: Thanks for your comment. We revised the text according to your comment.

Rotator Cuff injuries are a common cause of shoulder pain and disability.

(page 1, line 33-34)

Line 45-48. The incidence of deep infections has been previously reported between 0.03% and 3.4% representing a potentially devastating postoperative complication while the incidence of deep infections has been 47 previously reported between 0.03% and 3.4% [5].

→The incidence of deep infections has been previously reported between 0.03% and 3.4% representing a potentially devastating postoperative complication[5].

A: Thanks for your comment. We revised the text according to your comment.

The incidence of deep infections has been previously reported between 0.03% and 3.4% representing a potentially devastating postoperative complication.

(page 2, lines 48-49)

Line 153-155. “Cultures with no pathogens isolated amounted to 9.2% of the total. A total of 10 patients had multiple causative pathogens.” →The numbers don't add up. How many patients who don’t reveal the cultures?

A: Thanks for your comment. We revised the text and the Table 3 according to your comment.

Data for perioperative cultured pathogens were reported for 1431 cultures, and the most common causative bacteria were Cutibacterium acnes (50.446.2%), Staphylo-coccus epidermidis (17.59%), and coagulase-negative Staphylococcus (CNS) (14.613.3%) Cultures with no pathogens isolated amounted to 9.2% of the total. A total of 10 patients had multiple causative pathogens.

(page 6, lines 167-171).

Line 289-290. “Ammann et al[8]. found that the removal of anchors or sutures, repeated revision surgery, or antibiotic therapy beyond 6 weeks failed to improve remission or to reduce sequelae.” →Why? Please elaborate more detail.

A: Thanks for your comment. We revised the manuscript according to your comment providing the recommendation of Amman et al. in order to avoid implant removal and to not affect the success of the revision surgery in terms of shoulder functionality.

Ammann et al [8]. found that the removal of anchors or sutures removal, repeated re-vision surgery reoperation, or antibiotic therapy beyond 6 weeks did not enhance re-mission or decrease failed to improve remission or reduce sequelae; these findings are a practical conclusion, in line with the literature, and the authors' recommendation is to avoid the simultaneous removal of the implant devices, unless they are loose and easily surgically accessible.

(page 10, lines 318-323)

Line 302-304. “Atesok et al. [21] suggested that the initial intravenous antibiotic treatment can be then supplemented or extended by oral antibiotics depending on the causative microorganism and the patient’s response.” →Why? Please elaborate more detail.

A: Thanks for your comment. We revised the text according to your comment.

Atesok et al. [21] proposed that the initial intravenous antibiotic treatment, performed 4 to 6 weeks, is supplemented or prolonged by oral antibiotics depending on the causative phatogen and the patient’s response to be sure to resolve the infection. Assessment of therapeutic response and follow-up is mainly based on patient’s clinic and monitoring of infection markers. The duration of antibiotic treatment and confirmation of eradication of infection requires shared decision making between infectious disease specialists and the orthopaedic.

(page 10, lines 337-344)

Reviewer 2 Report

Comments and Suggestions for Authors

Dear Authors,

This study highlights a topic remaining hot as infection prevention and management is a constant fear and problem in orthopaedics. As far as I am considered, there are some points needing improvement and further clarification. The most significant is the lack of any analysis of surgical debridement and the absence of any mention regarding antibiotics (regimen, dosage). Even in heterogenous studies, this information should be included. 

Line 13: Please mention in short what is defined as "deep infection" in abstract section.

Lines 45-48: Please rephrase as this sentence makes no sense.

Line 71: Please mention if this study was registered prospectively or retrospectively in PROSPERO.

Line 73-75: Please provide the search strategy apart from terms in the Appendix.

Line 73: Please clarify the meaning of "published in January 2024". Exclusively in January or until January 2024? And provide the exact date.

Lines 81-84: Please provide further data regarding inclusion criteria. To be more exact, clarify infection onset (early - late) and mention the rationale of more than 5 participants studies included.

Lines 94-97: Please provide a short explanation of these scores (with reference).

Line 134 (Figure 1): The flowchart of your review has no additional records which seems peculiar. There were no clinical trials ongoing during this period about postoperative rotator cuff tear infections? Moreover, the number of 390 studies included after duplicates removed seems rather small. You should extend the search in order possible bias to be limited. 

Line 139 (Table 2): Please reform the table. The most important data is included in the latter columns where narrow space makes any apprehension chaotic. 

Line 150 (Table 3): Please add a column with antibiotic treatment if this is possible and provide data regarding the surgical technique used (open or arthroscopic) and the way of debridement. Which implants were removed? Moreover mention the exact germ where "other staphylococcus species" is written as it is only one and create wider groups summing all staph species, gram positive cocci etc. If the latter is impossible, please mention the rationale. The Methicillin-Resistant factor can be remained as it is.

Line 172 (table 5): You could add in this table a column regarding the technique of debridement. These excellent outcomes regarding the infection eradication is the result of a good surgical technique. So this should be mentioned.

Lines 191-192: Please provide dosage apart from duration if this information is available.

Lines 198-258: Infection prevention and risk factors are important but out of scope regarding this study. So, it is advisable to shorten these data into a single  paragraph.

Author Response

May 28th, 2024

Rahman Shiri                      

Editor-in-Chief

Healthcare

Manuscript ID Healthcare-2973045

Title: Revision surgery for shoulder infection after arthroscopic rotator cuff repair: functional outcomes and eradication rate – a systematic review

Dear Editor,

Enclosed please find our point-to-point response to the comments concerning our manuscript entitled “Revision surgery for shoulder infection after arthroscopic rotator cuff repair: functional outcomes and eradication rate – a systematic review” that is being considered for publication in your Journal.

Changes were made with word track changes. We hope that the revision improves our manuscript and that the work will be judged worthy of publication in your Journal.

Sincerely,

The Authors

Reviewer #2

This study highlights a topic remaining hot as infection prevention and management is a constant fear and problem in orthopaedics. As far as I am considered, there are some points needing improvement and further clarification. The most significant is the lack of any analysis of surgical debridement and the absence of any mention regarding antibiotics (regimen, dosage). Even in heterogenous studies, this information should be included.

 A: Thanks for your comment. We agree with the limitations of the study that you mentioned. We revised the text point-to-point according to your valuable suggestions as well as we reported in detail the limitations of the study.

Line 13: Please mention in short what is defined as "deep infection" in abstract section.

A: Thanks for your comment. We revised the text according to your comment.

The incidence of deep infections (defined as an infection involving any part of the anatomy other than the skin and subcutaneous tissue) after surgery ranges between 0.03% and 3.4%.

(page 1, lines 13-14)

Lines 45-48: Please rephrase as this sentence makes no sense.

A: Thanks for your comment. We revised the text according to your suggestion.

The incidence of deep infections has been previously reported between 0.03% and 3.4% representing a potentially devastating postoperative complication.

(page 2, lines 48-50)

Line 71: Please mention if this study was registered prospectively or retrospectively in PROSPERO.

A: Thanks for your comment. We revised the text according to your comment.

The study protocol was registered prospectively in PROSPERO (CRD525387).

(page 2, lines 76-77)

Line 73-75: Please provide the search strategy apart from terms in the Appendix

A: Thanks for your comment. We added the search strategy in the Appendix 1 at the end of the manuscript.

Line 73: Please clarify the meaning of "published in January 2024". Exclusively in January or until January 2024? And provide the exact date.

A: Thanks for your comment. We revised the text according to your comment.

The PubMed, MEDLINE, Scopus, and Cochrane Central databases were searched for articles published until January 2024.

(page 2, lines 77-78)

Lines 81-84: Please provide further data regarding inclusion criteria. To be more exact, clarify infection onset (early - late) and mention the rationale of more than 5 participants studies included.

A: Thanks for your comment. We revised the text according to your comment.

Population: patients with early and late infection after shoulder arthroscopy for rotator cuff repair; 2) Intervention: studies reporting >5 surgically cases, to avoid case report and case series until 4 patients, treated with open or arthroscopic surgery to manage infection.

(page 2, lines 90-92)

Lines 94-97: Please provide a short explanation of these scores (with reference).

A: Thanks for your comment. We revised the text according to your comment.

The data extracted for quantitative analysis included the timing of shoulder infection, the visual analogue scale (VAS) score for pain, ranging from 0 (no pain) to 10 (extreme pain)[10]. The American Shoulder and Elbow Surgeons (ASES) score contains a physician-rated and patient-rated score, evaluation of pain and 10 functional questions[2]. The Constant-Murley score (CMS), it comprises measures of capacity (ROM and strength) and subjective parameter as work load, pain, and leisure time activities with a score ranging from 0 (worst) to 100 (best) [2].

(page 3, lines 104-112)

Line 134 (Figure 1): The flowchart of your review has no additional records which seems peculiar. There were no clinical trials ongoing during this period about postoperative rotator cuff tear infections? Moreover, the number of 390 studies included after duplicates removed seems rather small. You should extend the search in order possible bias to be limited. 

A: Thanks for your comment. We performed a new search and provided the details of the search strategy in the Appendix 1. Moreover, we provided a new Figure 1 accordingly. We also performed a search of clinical trials ongoing without results.

Line 139 (Table 2): Please reform the table. The most important data is included in the latter columns where narrow space makes any apprehension chaotic.

A: Thanks for your comment. We revised the width of the columns of table 2.

Line 150 (Table 3): Please add a column with antibiotic treatment if this is possible and provide data regarding the surgical technique used (open or arthroscopic) and the way of debridement. Which implants were removed? Moreover mention the exact germ where "other staphylococcus species" is written as it is only one and create wider groups summing all staph species, gram positive cocci etc. If the latter is impossible, please mention the rationale. The Methicillin-Resistant factor can be remained as it is.

A: Thanks for your comment.

Regarding the type of surgical technique used it was specified in Table 5 and Table 6, dividing all the cases of the various studies in arthroscopic and open revision surgery. The way of debridement was not specified in the included studies so we could not report these results.

The type of antibiotic treatment was not specified in the included studies in terms of dosage and  type of antibiotics.

The implant removes were sutures and anchors. We provided this part in the “treatment options and eradication rate” section.

A total of 110 sutures and anchors in 123 patients were removed. (page 8, lines 188-189).

The only study that reported “Other Staphylococcus species” was Stone et al. but the authors did not specified the type of Staphylococcus.

Line 172 (table 5): You could add in this table a column regarding the technique of debridement. These excellent outcomes regarding the infection eradication is the result of a good surgical technique. So this should be mentioned.

A: Thanks for your comment. The included studies did not report details of the surgical technique used for the debridement. We provided this aspects in the limitations section.

Third, it was not possible to compare outcomes according to the type of revision surgery used, arthroscopic or open, and there was heterogeneity in terms of treatment of infection between the included articles that did not provide details of the surgical technique used to manage infection.

(page 10, lines 351-354)

Lines 191-192: Please provide dosage apart from duration if this information is available.

A: Thanks for your comment. The type of antibiotic treatment was not specified in the included studies in terms of dosage and  type of antibiotics.

Lines 198-258: Infection prevention and risk factors are important but out of scope regarding this study. So, it is advisable to shorten these data into a single paragraph.

A: Thanks for your comment. We shortened this paragraph as you suggested. Moreover, we previously provided this paragraph to reach the minimum number of words requested according to the editorial staff recommendation.

Round 2

Reviewer 1 Report

Comments and Suggestions for Authors

The author has corrected this paper throughout, as I have noted.

I think all tables could be changed to be more readable.

In table 3, GPC is include MSSA, Staphylococcus epidermidis, Staphylococcus aureus, and so on. Please correct this.

Reviewer 2 Report

Comments and Suggestions for Authors

Dear Authors,

You have made a honourable effort to improve the manuscript quality but further improvement needs to be done. Adding information and aspects of a study in limitation section is a clever technique but provides less in significance of the content. In case of high heterogeneity among the studies a narrative review is a possible solution. Primary or revision index surgery, open or arthroscopic debridement, number of re-debridement and possible alteration of the method, duration of the treatment and the follow-up, pre-and postoperative functional scores should be clearly presented. Some of the above are included in your study but in a complex fashion. 

Line 83: Please include in the Appendix only one version of your research strategy in only one database.

Lines 83-86: This sentence makes no sense.

Lines 86-89: Please reform the flowchart figure adding the additional records  from other sources as it mentioned in these lines.

Line 94: It is obvious that having as an inclusion criterium case series with more than 5 patients, any case series with less than 4 participants are excluded. I have asked to justify this number. Is it arbitrary or is there any rationale?

Line 151: There is no quantitive synthesis in this review as there is high heterogeneity among the studies. So, the last step of the flowchart is not applicable to the present review.

Line 170: I have already mention this : "Line 150 (Table 3): Please add a column with antibiotic treatment if this is possible and provide data regarding the surgical technique used (open or arthroscopic) and the way of debridement. Which implants were removed? Moreover mention the exact germ where "other staphylococcus species" is written as it is only one and create wider groups summing all staph species, gram positive cocci etc. If the latter is impossible, please mention the rationale. The Methicillin-Resistant factor can be remained as it is." Please address it in adequate fashion.

Line 206: Discussion section seems ok. Everything is appropriately addressed.

Line 332: There is a typographical error.

Round 3

Reviewer 2 Report

Comments and Suggestions for Authors

Dear authors,

As far as I am concerned, everything is adequately addressed. To reach excellent level I would like to be included this answer of yours into final version "The rationale is based on not including articles with a higher risk of bias that could disqualify the

systematic review. Indeed, it was reported that case reports that typically include up to 4 patients,

are limited by their retrospective and nonblinded design, constituting a source of bias that could

affect the study outcome. Moreover, findings provided by case reports, may not be generalizable

and cannot be useful to establish a cause-effect relationship as reported by Sampayo-Cordero et al.

(Sampayo-Cordero M, Miguel-Huguet B, Malfettone A, Pérez-García JM, Llombart-Cussac A,

Cortés J, Pardo A, Pérez-López J. The Value of Case Reports in Systematic Reviews from Rare

Diseases. The Example of Enzyme Replacement Therapy (ERT) in Patients with

Mucopolysaccharidosis Type II (MPS-II). Int J Environ Res Public Health. 2020 Sep

10;17(18):6590. doi: 10.3390/ijerph17186590. PMID: 32927819; PMCID: PMC7558586)"